# Implementation of the Dutch expertise centre for child abuse: descriptive data from the first 4 years

Rick Robert van Rijn,[1,2] Marjo J Affourtit,[3] Wouter A Karst,[2] Mascha Kamphuis,[4] Leonie C de Bock,[4] Elise van de Putte,[5] on behalf of the Dutch Expertise Centre for Child Abuse Study Group

[1]Radiology, Emma Children's hospital, Academic Medical Center, Amsterdam, The Netherlands
[2]Forensic Medicine, Nederlands Forensisch Instituut, Den Haag, The Netherlands
[3]Pediatrics, Erasmus MC Sophia Children's Hospital, Rotterdam, The Netherlands
[4]Dutch Expertise Center for Child Abuse, Utrecht, The Netherlands
[5]General Paediatrics, University Medical Center Utrecht, Utrecht, The Netherlands

**Correspondence to**
Dr Rick Robert van Rijn;
r.r.vanrijn@amc.uva.nl

## ABSTRACT

**Objective** Combined paediatric and forensic medical expertise to interpret physical findings is not available in Dutch healthcare facilities. The Dutch Expertise Centre for Child Abuse (DECCA) was founded in the conviction that this combination is essential in assessing potential physical child abuse. DECCA is a collaboration between the three paediatric hospitals and the Netherlands Forensic Institute. DECCA works with Bayes' theorem and uses likelihood ratios in their conclusions.

**Design** We present the implementation process of DECCA and cross-sectional data of the first 4 years.

**Participants** Between 14 December 2014 and 31 December 2018, a total of 761 advisory requests were referred, all of which were included in this study. An advisee evaluation over the year 2015 was performed using a self-constructed survey to gain insight in the first experiences with DECCA.

**Results** 761 cases were included, 381 (50.1%) boys and 361 (47.4%) girls (19 cases (2.5%) sex undisclosed). Median age was 1.5 years (range 1 day to 20 years). Paediatricians (53.1%) and child safeguarding doctors (21.9%) most frequently contacted DECCA. The two most common reasons for referral were presence of injury/ skin lesions (n=592) and clinical history inconsistent with findings (n=145). The most common injuries were bruises (264) and non-skull fractures (166). Outcome of DECCA evaluation was almost certainly no or improbable child abuse in 35.7%; child abuse likely or almost certain in 24.3%, and unclear in 12%. The advisee evaluations (response rate 50%) showed that 93% experienced added value and that 100% were (very) satisfied with the advice.

**Conclusion** Data show growing interest in the expertise of DECCA through the years. DECCA seems to be a valuable addition to Dutch child protection, since advisee value the service and outcome of DECCA evaluations. In almost half of the cases, DECCA concluded that child abuse could not be substantiated.

## INTRODUCTION

Child abuse is a worldwide ubiquitous problem leading to both short-term and long-term negative outcomes in health and social well-being of the victims.[1 2] It has a detrimental impact on the development of children, with an impact on the individual

### Strengths and limitations of this study

► After 4 years of advisory requests, we have collected a large number of cases to show solid results. With the growing number of advisory requests and satisfied advisees, we were able to show that organising and providing round-the-clock, expert-based independent high-quality combined paediatric and forensic medical expertise on a national level is feasible.

► During the initial discussion between the paediatrician, who has spoken with the advisee, and the forensic physician, the latter is blinded with respect to the social history and other risk factors. This way we aim to avoid cognitive bias as much as possible.

► As the Dutch Expertise Centre for Child Abuse works with anonymous advice, follow-up of the individual cases is not possible. Therefore, a health economics analysis, although important, is impossible.

► The results of the advisee evaluations could be biased, because possibly only satisfied advisees responded to the survey.

► An update of the advisee evaluations is needed and this has been implemented as a routine part of our protocol since 1 January 2019.

throughout life. The seminal study by Felitti et al has shown that there is an increase of disease and early death in adulthood in child abuse survivors.[1] Besides the direct impact on the victim, it has been shown to have a significant cost to society.[3 4]

Both from an individual as well as a societal point of view, it is important to prevent and detect child abuse as early as possible and intervene when child abuse is suspected in order to prevent further abuse and to improve the outcome of the individual victim and his/her social network.

In the Netherlands, in contrast to other European countries and the USA, physicians are only mandated to report child abuse in specific cases according to the guideline domestic violence and child abuse

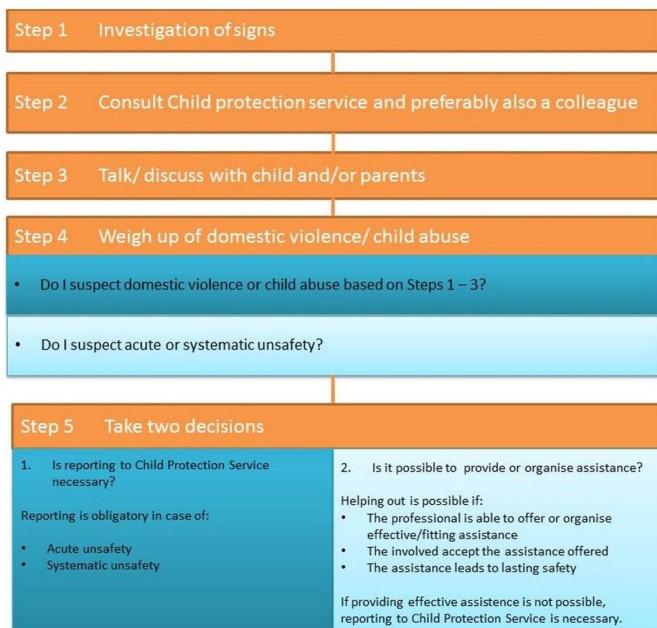

**Figure 1** Stepwise approach of managing child abuse as guideline published by the Royal Dutch medical Association.[5]

as published by the Royal Dutch Medical Association (figure 1).[5] This guideline dictates five steps, of which the first step is to collect information about the patient in question and assess the potential for the presence of child abuse. If the attending physician thinks that child abuse might be present, he/she is required to ask advice from or report to so-called child safeguarding doctors (step 2). Child safeguarding doctors work regionally within a national system, in which all cases of possible child abuse are reported and investigated and they can, in collaboration with a wide variety of healthcare support teams, instigate interventions.

However, in order to substantiate potential abuse as well as reject this diagnosis in the first step of the diagnostic pathway, specifically concerning injuries on physical examination and questions concerning physical signs and symptoms, in-depth knowledge of both paediatric as well as forensic medicine is required. We will refer to this combination of expertise as combined paediatric forensic medical expertise. In addition, other relevant subspecialties and affiliated specialties, such as radiology, will add to this knowledge. This adheres to the recommendation of WHO Regional Office for Europe report on the prevention of child maltreatment, to use a multidisciplinary approach of cases that use reliable and valid investigative methods (2013).[6] In the Netherlands, as in most countries, this broad combined range of specialties is not available in hospitals or healthcare facilities. Given this fact and the necessity for 24/7 support for healthcare professionals confronted with a potential case of child abuse, we founded the 'Landelijk Expertise Centrum Kindermishandeling' (Dutch Expertise Centre for Child Abuse (DECCA)) in 2014.[7] DECCA can be consulted in step 1, step 2 and step 5 of the national guideline domestic

violence and child abuse.[5] Child death reviews are not part of the objectives of DECCA. In this publication we present the organisation and methods of the DECCA and our initial experiences in the first 4 years.

## Organisation of the DECCA
Advice and consultations are performed by DECCA-paediatricians and DECCA-forensic physicians. The DECCA is a tax-exempted foundation initiated by the three university paediatric hospitals (Emma Children's Hospital— Amsterdam UMC—University of Amsterdam, Sophia Children's Hospital— Erasmus MC Rotterdam and Wilhelmina Children's Hospital—UMC Utrecht) and the department of forensic medicine of the Netherlands Forensic Institute, The Hague. The DECCA as such is a virtual organisation: it has no buildings. DECCA is supervised by a board (consisting of a physician from each centre and a treasurer), a central coordinator, a secretary and since 2018 a medical director. In each centre, one of the involved physicians is in the lead and these four team leaders meet on a semiannual basis with the board of DECCA to discuss points for improvement and future developments. In addition, an advisory board and advisors meet up with the board two to three times a year.

## Method of DECCA
The DECCA is available for healthcare professionals 7 days a week, day and night, by a national telephone number that is staffed by a paediatrician with expertise and experience in child abuse diagnostics (an additional 2½ years of education). At the same time a forensic physician, with expertise and experience in child abuse evaluation, is on call. Specific quality criteria for the physicians working at DECCA are described and are maintained by the board and medical director of DECCA.

Advisees of the DECCA are mainly medical doctors who ask for a one-off advice (by telephone or email) or refer for consultation. Besides medical information, photographs and radiography can be provided for review. Focus is mainly on physical injury and questions concerning physical signs and symptoms. For all cases where advice is requested DECCA records the patient's data anonymously. Because of this, DECCA has a low threshold for physicians, as in the Netherlands anonymous advice can be asked without permission from parents or caregivers.

In all cases, the advisee will get a combined advice from both a paediatrician and a forensic physician. During the initial discussion between the paediatrician, who has spoken with the advisee, and the forensic physician, the latter is blinded with respect to the social history and other risk factors. This way we aim to avoid cognitive bias as much as possible. At first, a conclusion is formed using a likelihood ratio (ie, the likelihood that the medical findings or injuries would be expected in an abused child compared with a non-abused child). This conclusion is substantiated with data from scientific publications. To determine the likelihood of abuse, all other information (apart from the injuries) should be used to estimate a

prior probability of abuse. Hence, the injuries should not be used to determine the prior probability of abuse, they should only be used to determine the likelihood ratio. Using Bayesian reasoning, the prior odds of abuse can be multiplied by the likelihood ratio that was determined by the forensic physician who was blinded for risk factors of abuse. This way, cognitive bias will be avoided as much as possible. As the information needed to determine the prior probability of abuse might be incomplete, subjective, unknown or outside the field of expertise of medical doctors, a correct (posterior) probability of abuse cannot be given by DECCA. If a probability is needed to determine further action or evaluation, the prevalence of abuse could be used as prior probability. If DECCA provides prevalence data in specific situations, it is always stated that using those data to determine the probability of abuse might be the best solution, but also might be incorrect.

The search for evidence and the discussion between the paediatrician and the forensic physician continues until they agree. In case of different opinions or lack of evidence, additional advice is obtained from other experts both nationally as well as internationally. In all cases in which DECCA is asked for advice, the advisee will receive a letter stating the anonymous information provided by the advisee, the conclusions reached by DECCA and if needed the advice for further assessment or follow-up. It is specifically stated that this letter should be included in the patient's medical record. All cases are recorded in a secure online data base (Castor EDC, CIWIT B.V., Amsterdam, the Netherlands) to which all DECCA professionals have password protected access to, including two-factor authentication. The database conforms to the guidelines of Good Clinical Practice and the General Data Protection Regulation (GDPR).[8] Advice will be given anonymously and personal information of the child or the family is not added to our data base. Name and email address of the advisee is included in the data base, in order to be able to send the letter of advice to the advisee.

A DECCA case-review telephone conference in which all centres are engaged and at least two senior paediatricians and a forensic physician must be present is held weekly. During this telephone conference, all cases of the past week are discussed anonymously. In this discussion, the paediatrician who initially handled the case presents it to the participating DECCA physicians and the advice given is discussed, including the written report. Based on the group discussion, the advice can be altered. In case of an updated or altered advice, the referring healthcare professional is contacted to discuss this update. The DECCA is not involved in further treatment or follow-up of the cases.

In those cases in which additional specific expertise for physical examination or additional investigations is required, DECCA can advise referral of the patient to a paediatric centre with expertise on child abuse (including one of the three centres collaborating in the DECCA) for outpatient follow-up and evaluation. For example, this could be the case in investigation of signs of sexual abuse. On average, 35 cases per year are referred to one of the three academic paediatric hospitals of DECCA for face-to-face consultation. These cases are part of academic patient care, hence not the main focus of DECCA, and are therefore not presented in this overview.

## METHODS: DATA COLLECTION AND STATISTICAL ANALYSES

Between 14 December 2014 and 31 December 2018 a total of 761 advisory requests were recorded by the DECCA paediatricians in the DECCA database.

Additionally, an evaluation of the experiences of advisees was performed. This survey was specifically developed for this study and is available (in Dutch) on request. All advisees that consulted DECCA from December 2014 to December 2015 received an online survey consisting of 15 questions about their advisory request. This related to how they experienced the added value of DECCA, if they were satisfied with the advice as such, and how DECCA could improve their service. This evaluation was done from May 2015 to January 2016. During this time, three reminders were sent. In total 122 advisees were asked to participate anonymously in an online questionnaire sent directly after they had received advice.

Statistical analysis was performed using IBM SPSS Statistics for Windows, V.25. (IBM). Analyses involved the processing of frequencies or cross-tables.

## PATIENT AND PUBLIC INVOLVEMENT

As DECCA does not record patient identifiers, patients and/or their parents could not be included in this descriptive study of the first 4 years of DECCA.

## RESULTS

### Advisory requests

All cases of advisory requests recorded in the data base were included in the analyses, this concerns a total of 761. Figure 2 shows the increasing number of requests during the years. There were 381 (50.1%) boys and 361 (47.4%) girls; in 19 cases (2.5%) sex was not disclosed. Age was not recorded in nine cases. The median age of the referred population was 1.5 years (IQR 3.67, range 1 day to 20 years). The oldest case was a 20-year-old patient who, although too old according to the definition of child abuse, was a sibling of a younger child and was therefore referred to DECCA and included in this study. The median age in boys was 1.4 years (IQR 3.06) and in girls this was 2 years (IQR 5.35). The majority of cases was referred from the Western provinces of the Netherlands, corresponding to a more densely populated area (figure 3). The cases were referred throughout the week, though mainly during weekdays (91.9%). Cases were referred throughout the day and night, however mostly during office hours (75.9%).

In 2018, 229 advice requests were evaluated by LECK

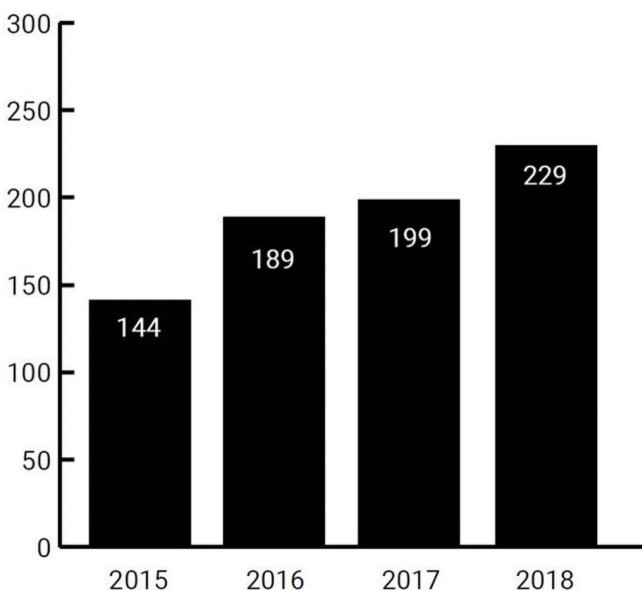

**Figure 2** Number of DECCA advisory requests from 2015 to 2018. DECCA, Dutch Expertise Centre for Child Abuse.

There was a wide variety of healthcare professionals who consulted the DECCA for advice on cases in which they had findings concerning (suspected) child abuse (table 1). Paediatricians were the most prevalent advisees (53.1%), with child safeguarding doctors coming second. This is shifting over the years towards relatively more requests from child safeguarding doctors (from 21% in 2015 to 26% in 2018) and relatively less from paediatricians (from 59% in 2015 to 52% in 2018) (table 1).

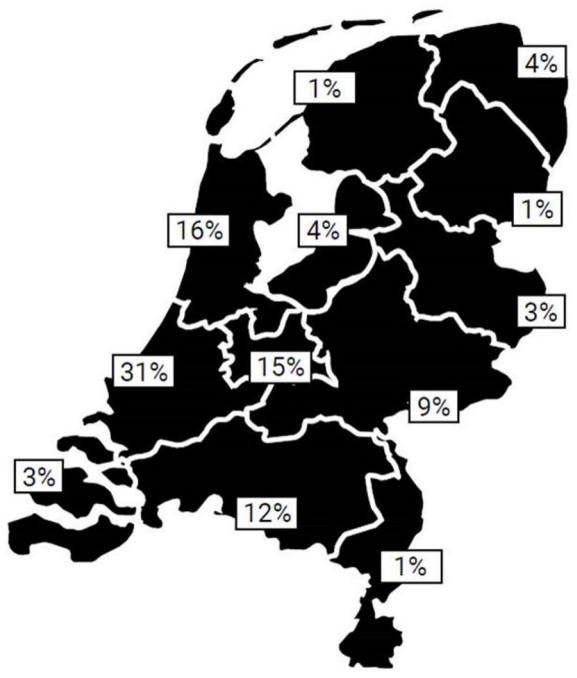

**Figure 3** Geographical distribution of the advisee to the Dutch Expertise Centre for Child Abuse (N=740, 21 missing).

**Table 1** Overview of advisees to DECCA (n=761)

| Specialty | N | % of all 761 cases |
|---|---|---|
| Paediatrician | 404 | 53.1 |
| Child safeguarding doctors | 167 | 21.9 |
| General practitioner | 20 | 2.6 |
| Youth healthcare doctor | 12 | 1.6 |
| Emergency room doctor | 9 | 1.2 |
| Youth care worker | 7 | 0.9 |
| Others (like surgeons, general practitioners or paediatricians in training, psychiatrist, dentist, forensic nurses at ER, nurse practitioners) | 142 | 18.7 |

DECCA, Dutch Expertise Centre for Child Abuse; ER, Emergency Room.

In some cases, expertise from other medical specialists was needed, DECCA, therefore, involved several clinical subspecialties in the advice process besides paediatrics and forensics, primarily paediatric radiology (table 2).

The primary question of the advisee mainly concerned 'Does the observed injury fit the described trauma mechanism?' and 'Could the nature of the injury be ascertained?'. As is common in (suspected) child abuse, there were often more than one finding or so-called red flags leading to a request for DECCA advice (table 3). Most common injuries in advisory requests were bruises (264), non-skull fractures (166), skull fractures (69) and brain injury (64) (table 4).

In addition to answering the specific question of the advisee and when possible, the conclusion of DECCA advice specifically states a level of certainty with respect to child abuse. Table 5 shows the outcome regarding the possibility of child abuse as concluded by DECCA. This

**Table 2** Subspecialties involved in DECCA (n=761)

| Subspecialty | N* | % of all 761 cases |
|---|---|---|
| Paediatric radiology | 296 | 38.9 |
| Paediatric dermatology | 31 | 4.1 |
| Paediatric ophthalmology | 22 | 2.9 |
| Paediatric neurology | 12 | 1.6 |
| Paediatric haematology | 7 | 0.9 |
| Social work | 7 | 0.9 |
| Paediatric surgery | 3 | 0.4 |
| Others (like paediatric urology, ear-nose-throat doctor, paediatric immunology) | 15 | 2.0 |

*The total does not sum up to 761 as not for every case consultant of a subspecialty is needed and several subspecialties can be consulted in one single case.
DECCA, Dutch Expertise Centre for Child Abuse.

van Rijn RR, *et al. BMJ Open* 2019;9:e031008. doi:10.1136/bmjopen-2019-031008

**Table 3** Findings or red flags leading to a request for DECCA advice

| Finding or red flag | N* |
|---|---|
| Injury/skin lesion | 592 |
| Clinical history not in keeping with findings | 145 |
| Presence of risk factors in family | 70 |
| Physical symptoms | 71 |
| Inconsistent clinical history | 47 |
| Lesion/trauma not in keeping with child's age | 40 |
| Delay in presentation | 39 |
| Previous trauma | 33 |
| Child admits being a victim of abuse | 26 |
| Behavioural symptoms child | 19 |
| Caregiver admits to child abuse | 17 |
| Improper hygiene child | 9 |
| Inadequate interaction child—caregiver | 8 |
| Other child in family discloses about child abuse | 8 |
| Findings at additional investigations (urine or blood test) | 2 |

*The total does not sum up to 761 as several findings or red flags can be present in one single case and several findings can be mentioned in one single case.
DECCA, Dutch Expertise Centre for Child Abuse.

was almost certainly no or improbable child abuse in 35.7%; child abuse possible in 28%; and child abuse likely or almost certain in 24.3%. In 12% of cases, additional investigations were advised.

### Advisee evaluation

In total 61 out of 122 (50%) advisees responded to the online evaluation questionnaire. Of these respondents 39 (64%) stated that this was the first time they contacted DECCA and the rest (36%) stated that they had previously been in contact with DECCA. In the majority of cases, the respondents found that the DECCA advice had either much additional value (N=51 (84%)) or very much additional value (N=8 (13%)). Only two respondents (3%) found that the DECCA advice was of little additional value. The DECCA advice changed the perspective on the case in 33 (54%) of cases. In all cases, the respondents were satisfied or very satisfied about the DECCA advice

**Table 4** Four most common injuries in advisory requests

| Most common mentioned type of injury | N* |
|---|---|
| Bruises | 264 |
| Non-skull fracture | 166 |
| Skull fracture | 69 |
| Brain damage | 64 |

*The total does not sum up to 761 as several findings or red flags can be present in one single case and not all injuries are being listed here.

**Table 5** Outcome of DECCA evaluation (N=753, 8 missing)

| Outcome of DECCA evaluation | No | % of 753 cases |
|---|---|---|
| Almost certainly no child abuse | 84 | 11.1 |
| Child abuse unlikely | 185 | 24.6 |
| Child abuse possible | 211 | 28.0 |
| Child abuse likely | 114 | 15.1 |
| Almost certainly child abuse | 69 | 9.2 |
| Unclear, more investigation needed | 90 | 12.0 |

DECCA, Dutch Expertise Centre for Child Abuse.

and marked it on average as 8.4 on a scale from 0 to 10 (range from 7 to 10, with 16 respondents scoring 9 and 5 scoring 10).

### DISCUSSION

Interest for contacting DECCA is growing nationally, as is shown by the increasing numbers of advisory requests (from 132 in 2015 to 229 in 2018). In the past 4 years, DECCA has shown that organising and providing expert-based and independent combined paediatric and forensic medical expertise on a national level is feasible. From both a care providers' as well as the child's viewpoint, DECCA's involvement had an impact on the outcome on the next steps in the reporting process of (suspected) child abuse, because, as was shown by the advisee evaluation, it added to, changed or confirmed the former conclusion of the advisee.

Teamwork in child abuse diagnostics in itself is not a novel development. In many countries, the hospitals have some form of a hospital-based multidisciplinary team, and these have proven their value over time.[9–12] Besides these hospital-based teams, many countries, as well as The Netherlands, have initiated regional or citywide rape and sexual abuse counselling centres, where in many cases specially trained sexual assault nurse examiners and/or child abuse paediatricians work.[13–15] These centres play an important role in the diagnosis of sexual child abuse and the care for the victims thereof. The specific aspect of DECCA is the combination of both paediatric and forensic knowledge, outside the field of justice, continuously available on a nationwide scale. Each specialty brings their own expertise. The paediatrician's main focus is on the medical differential diagnosis, whereas the forensic physician adds an objective assessment of the injury and the potential causative mechanism. In addition, the use of Bayes' theorem adds to the perspective of the advisee. This combination of expertise and method yields an integral analysis of the patient and his/her injury from two different viewpoints leading to an objective and evidence based analysis of suspected child abuse, with reference to relevant literature. To our knowledge, this makes DECCA unique in the world.

It has been shown that expert consultation has added value to patient care in cases of suspected child abuse.[10 16 17] However, Lindberg *et al* showed that even between child abuse paediatricians there was a significant variability in assessing the likelihood of child abuse.[18] They concluded that their data supported the use of peer-review or multidisciplinary teams. The weekly telephone conference, in which all DECCA cases are discussed in-depth, clearly addresses this issue. Not only does this provide an excellent opportunity to share knowledge and learn from one another, but more importantly, we strongly believe that our approach increases the quality and consistency of the advice given to the referrer.

A limitation of this study is that we could not do a health economics analysis. Long-term follow-up of (a subset of) cases will be extremely difficult especially since we register our cases anonymously. With the introduction of the GDPR, follow-up would only be possible with consent from both caregivers/parents/legal guardians. This would almost certainly introduce an inclusion bias. No inter-rater reliability assessment is done. However, we restructured our database and as of 2019 we register inconsistencies between the first advice and the joint assessment during the weekly peer review. This will eventually give us insight in potential shortcomings in our DECCA advice. In addition, the advisee evaluation has been restructured where advisees are requested to fill out a short questionnaire after each consultation. These two procedural changes will form the basis for a new quality assessment.

A major difficulty of diagnosing child abuse lies in one of the principal precepts of bioethics we teach all our medical students: 'primum non nocere'. In cases of potential child abuse, the clinician must weigh the risk of under-reporting versus over-reporting, as both can have serious consequences for the child and its caregivers. As a result it is well known that, even in countries with mandated reporting, there is a significant level of under-reporting of child abuse and neglect. One of the explanations for under-reporting is reported to be a lack of knowledge leading to uncertainty of the diagnosis of child abuse.[19–21] It could be argued that if expertise centres like DECCA are more widely available this could lead to an increase in diagnosing and reporting child abuse, but also in rejecting the diagnosis child abuse when not justified in an earlier phase of the diagnostic process.

The data we present have a significant drawback. As DECCA acts in the first or second step of the guideline as published by the Royal Dutch Medical Association and mainly focusses on findings during the physical examination, there is a distinct referral bias. As a result, cases where physical or sexual child abuse are suspected are over-represented in our study population in comparison to other child abuse studies. The role of a national paediatric forensic medical expertise centre will in most cases be limited to these two types of child abuse, where physical injury and physical signs are most evident. For cases in which neglect or deprivation play an important role, without signs of physical injury, a face-to-face clinical consultation with a paediatrician and/or child psychologist with expertise in this field remains the best solution. The strong involvement of three paediatric university hospitals guarantees that in forthcoming cases this service can also be offered to referring healthcare professionals.

The main challenge DECCA has faced was how to get its work financed. As DECCA works with anonymous cases, it, therefore, cannot bill the patient's insurance company. A grant from two Dutch charity funds, Stichting Kinderpostzegels and Nationale Postcode Loterij, made it possible to start DECCA in 2014 and paid for the running costs for the first year. Currently, the Dutch Ministry of Health, Welfare and Sport fully finances our work on a yearly basis. There are no studies into the economic added value of services like DECCA. However, a study into the value of telephone consultations of paediatric subspecialists to primary care physicians was shown to be cost-effective as a result of reduced use of costly services and reported improvements in quality of care.[22] We firmly believe that DECCA has a similar impact on patient care and being the sole provider of this care in the Netherlands, DECCA should be financed on a long-term project basis. This financial issue will play a role in other countries as well and could limit the implementation of similar models elsewhere.

## CONCLUSION

Our data show a rising number of advisory requests and satisfied advisees. We, therefore, conclude that DECCA seems to be a valuable addition to the Dutch system of child protection.

By virtue of its virtual centralised design this model could, in an adapted form, be established in other countries as well. We are convinced that the combination of experts in the field of paediatrics and forensic medicine including working with Bayes' theorem, and working together with affiliated (sub-)specialties, enables to provide available, evidence-based paediatric forensic medical expertise.

**Collaborators** Collaborators of the Dutch Expertise Centre for Child Abuse Study Group are:Department of Forensic Medicine, Netherlands Forensic Institute, the Hague, the Netherlands. Collaborators:Rob A.C. BiloHuub G.T. NijsHeike C. TerlingenSelina de VriesNicole L. van Woerden Department of Paediatrics, Emma Children's Hospital—Amsterdam UMC, University of Amsterdam, the Netherlands, Collaborators:Leonie van der Berg Annemarie C.M. van BellegemMachtelt BoumanMarie-Louise H. LoosAnika S. SmeijersAriane H. TeeuwDepartment of Paediatrics, Sophia Children's Hospital—ErasmusMC, Rotterdam, the Netherlands. Collaborators:Patrycja J. PuimanDepartment of Paediatrics, Wilhelmina Children's Hospital—UMC Utrecht, Utrecht, the Netherlands. Collaborators:Frederique M.C. van BerkestijnFemke KambergSanne L. NijhofJopje M. RuskampIngrid M.B. Russel-KampschoerMaartje SchoutenKarlijn SijstermansSaskia A.A. Wolt-Plompen.

**Contributors** RRvR collected data, carried out the initial analyses and drafted the initial and revised manuscript, and reviewed and revised the manuscript. MJA designed the data collection instruments and critically reviewed the manuscript. WAK designed the data collection instruments and critically reviewed the manuscript. MK carried out the secondary analyses, and drafted the revised manuscript, and critically reviewed and revised the manuscript. LCdB collected

data, carried out the secondary analyses and critically reviewed the manuscript. EvdP designed the data collection instruments and coordinated and supervised data collection, and critically reviewed the manuscript. All authors approved the final manuscript as submitted and agreed to be accountable for all aspects of the work.

**Funding** The Dutch Expertise Centre for Child Abuse is supported by grants from Stichting Kinderpostzegels, Nationale Postcode Loterij, Janivo Stichting, SOD and Fie van der Hoop fonds and governmental support from the Ministry of Health, Welfare and Sport.

**Map disclaimer** The depiction of boundaries on the map(s) in this article do not imply the expression of any opinion whatsoever on the part of BMJ (or any member of its group) concerning the legal status of any country, territory, jurisdiction or area or of its authorities. The map(s) are provided without any warranty of any kind, either express or implied.

**Competing interests** None declared.

**Patient consent for publication** Not required.

**Ethics approval** The internal review board of the University Medical Centre Utrecht, where DECCA has a postal address, issued a waiver for the documentation of consent and approved the use of an anonymised database.

**Provenance and peer review** Not commissioned; externally peer reviewed.

**Data availability statement** Data are available on reasonable request. No data are available.

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
