## [Reviewer comments · BMJ Open]

ARTICLE DETAILS

TITLE (PROVISIONAL)	Implementation of the Dutch Expertise Centre for Child Abuse: descriptive data from the first four years
AUTHORS	van Rijn, Rick; Affourtit, Marjo; Karst, Wouter; Kamphuis, Mascha; de Bock, Leonie; van de Putte, Elise

VERSION 1 – REVIEW

REVIEWER	Daniel Lindberg University of Colorado School of Medicine Colorado, USA
REVIEW RETURNED	07-May-2019

GENERAL COMMENTS	Thank you for the opportunity to review this revision. The authors have made substantial improvements, but I still have several concerns. MAJOR CONCERNS: - Please report the ethics approval process. I suspect this may be considered exempt from review, but please state this.- The authors lean heavily on their use of Bayes theorem, but give little information about what this means. It is not sufficient to simply state that "Bayes theorem was applied" because this will mean different things to different providers. I can imagine someone thinking that the best way to use Bayes theorem might be to say that the pre-test odds of physical abuse in all infants with femur fractures is 1:2, and that a transverse fracture conformation carries an LR+ of 0.9, and that the presence of a co-occurring tibia fracture has an LR+ of 4.5, so the post-test odds are 2:1, and this is therefore a case where "Child abuse is likely", without asking if the caregiver reported abuse, or if the child had been witnessed to be injured after a stairway fall while carried by their parent, who landed on their leg. There are many other ways to apply Bayes' theorem that would be better, but a reader can't evaluate the use of Bayes' theorem with the information provided. Please give an example of how Bayes' theorem is included in the advice given to referring doctors.- The abstract is not sufficient to summarize the piece. Please include in the abstract that DECCA focuses on physical injuries. Please add more methods to the abstract: inclusion and exclusion criteria, the methods by which outcomes are measured (the abuse likelihood scale, how it was assigned, how satisfaction was measured) enrollment periods, survey methods - when/how/to whom was the survey distributed and how was a response encouraged?- Survey methods are inadequate. Please include the complete survey as an appendix. Please report survey methods (above). How was it determined that the DECCA advice "changed the
---

	perspective on the case"? What is the threshold for a changed perspective?  - Either give the data, or don't report the result. If you say that cases were referred "mainly during weekdays" or "mostly during office hours", you need to give numbers to back this up. Same for "...shifting over the years towards more requests from child safeguarding doctors." MINOR CONCERNS  - The authors' written English is certainly superior to my written Dutch - but the manuscript will require some extensive proof-reading for grammar. - If 51.3% were boys, it is redundant to say that 48.7% were girls (but these percentages seem to exclude the 19 participants where sex was unknown). - As I wrote before, the statistics in the paper are not well-chosen. The authors still report both median and mean for the same number. Rather than reporting every possible measure of central tendency and spread, the authors should choose the correct measure and report that one. For normally distributed data, this is the mean and standard deviation. For other data, this is generally the median and IQR. - Why the parentheses in the phrase "100% was (very) satisfied..." - The conclusions should summarize, rather than re-capitulate the results. - Strengths and limitations continue to report results, rather than report the *methodological* strengths and limitations of the study. Examples might include: Strengths - rather large sample with 4 years of data to show change. Limitations - relatively low survey response rate could be vulnerable to response bias (i.e., the least satisfied referrers might be least likely to fill out the survey - especially if not anonymous). - I suggest that you search your document for the word "optimal" and replace it in every instance. To me, "optimal" implies - the best possible with no possibility of improvement. This is difficult to prove, and is not supported by these data. Similarly "despite its undeniable value" also seems to over-state the case. - When you say that the judgement of the injury is independent of social history, how can you ensure this? Keenan has shown implicit bias resulting from social data. Are experts blinded to social history? Is this even appropriate? After all, if you show me 2 children with identical fractures and tell me that one of them lives in a home with another abused child, I would reasonably believe that the risk of abuse is higher in that child. - I would not say (page 24) that geographic distribution was "skewed". Referrals came from the population centers, which is appropriate. - When you report that the most common injuries were "haematoma" - does this mean a bruise? subcutaneous hematoma? retroperitoneal hematoma? subdural?
--	--

REVIEWER	Gabriel Otterman Uppsala University Children's Hospital, Sweden
REVIEW RETURNED	12-May-2019

GENERAL COMMENTS	Thank you for inviting me to review your manuscript entitled "The start of the Dutch Expertise Centre for Child Abuse; a descriptive study of the implementation and data of the first 4 years." I refer to my comments and author responses to reviewer and editorial comments to the submission from a year ago "bmjopen-2018-
--

023608.” The authors have included these discussions and responses in the current submission and have revised the manuscript with inclusion of an additional year of data as well as a helpful STROBE checklist.

The authors have carefully and systematically responded to address issues brought up in the comments to the 2018 submission in a way that has improved the manuscript to my satisfaction. In this renewed review I provide some suggested minor revisions for the authors to consider in order to further improve the manuscript for publication.

The authors have consulted a native English speaker for correction of minor spelling and grammar issues to consistently follow the Journal’s style in UK English. The current manuscript contains some persistent style and grammar errors. A careful edit by a professional UK English editor is recommended. An example is the new title, where “start of” and “implementation” are perhaps redundant. Consider for example – “Implementation of the Dutch Expertise Centre for Child Abuse: descriptive data from the first four years.” Other examples are use of “advices” in plural, inconsistent spelling of “hematoma / haematologist.”

In reference to the question by another peer review regarding the level of care (p10) that is not routinely recorded by DECCA, do the authors not consider this pertinent data? I’m thinking in particular about validated prediction tools for abusive head trauma evaluations such as PediBIRN which is correctly applied to the ICU population. Certainly, level of neurological impairment is an important data point for assigning likelihood ratios for this diagnosis. Given the low median age of DECCA referrals, abusive head trauma should be a common concern in this group (brain damage - Table 4).

Despite the clarifications made by the authors in the new manuscript, I am still not clear on why the consultations with DECCA are anonymous. The authors cite confidentiality constraints such as GDPR as presenting a challenge to follow up of individual cases, as this would require obtaining consent from the legal guardians / parents. I would argue that complete quality improvement cannot be secured without having the ability to follow up cases and outcomes of inter-agency investigations, child protective services and criminal justice over time.

There are clearly some exceptions in which DECCA recommends referral to an appropriate centre with expertise (p 9, line 18). When DECCA provides a letter with an assigned likelihood of an abuse diagnosis, it is stated by DECCA that the letter should be included in the patient’s medical record (p 9, line 1). Are there not regulations that require the referring physician to obtain consent for the anonymized consultation? Is the letter provided by DECCA documented in the medical record not made available to the patient/parent/guardian? If so, is not DECCA accountable to supervisory medical authorities as medical providers who have provided opinion on the record? Are DECCA providers not brought in to child protective or criminal justice investigations based on the provided opinion?

An anonymised DECCA consultation letter may be illustrative if provided as supplemental material for this article.

	I seek further clarification on these issues as it is a matter that we struggle with in our practice setting (Sweden) where we also are compelled to comply with GDPR regulations. It is my opinion that anonymising the consultation reduces the ability to ensure systematic improvement, and obstructs accountability in the system. These issues may thereby limit scalability of DECCA across countries and practice settings. The authors state that inter-rater reliability of DECCA consultations has not been assessed, and yet assert that an advantage of the system is the assignment of a likelihood ratio for abuse versus non-abuse in accordance with a Bayesian analysis. Should not a quantified LR provide a ready means for inter-rater reliability? These could be assessed with the use of constructed case vignettes or anonymised cases. The investigators state that the DECCA final advice provided in the consultation is “independent of social history” (p8, line 20). I understand that the experts may be striving to avoid cognitive biases that may be introduced through elements of social history. This does not however acknowledge the fact that referrals to DECCA are often triggered by the presence of social history concerns such as “presence of risk factors in family” (Table 3). I would argue that you cannot “unsee” the reason for the referral as a component that has bearing on the base rate for abuse in a Bayesian analysis. The authors risk overstating the case for DECCA by characterising it as “unique” (p 13, line 8, 16) or “optimal” expertise (p 16, line 13). These attributes have not been supported by the data. Finally, I’d like to restate that there is a need for programmes such as DECCA, especially as many paediatric colleagues, especially across Europe, seek to find ways to access reliable expertise in an expedited manner. A revised manuscript should be of compelling interest to many readers of BMJ Open.
--	--

VERSION 1 – AUTHOR RESPONSE

Reviewer: 1

Name: Daniel Lindberg

Institution: University of Colorado School of Medicine, Colorado, USA
Competing interests: I have no competing interests

Please leave your comments for the authors below. Thank you for the opportunity to review this revision. The authors have made substantial improvements, but I still have several concerns.

MAJOR CONCERNS:

- Please report the ethics approval process. I suspect this may be considered exempt from review, but please state this.
 - We have added a paragraph on the Ethical approval. As the reviewer correctly suspected the internal review board of the University Medical Centre Utrecht, where DECCA has its postal address, has issued a waiver for the documentation of consent and the use of an anonymized database.

- The authors lean heavily on their use of Bayes theorem, but give little information about what this means. It is not sufficient to simply state that "Bayes theorem was applied" because this will mean different things to different providers. I can imagine someone thinking that the best way to use Bayes theorem might be to say that the pre-test odds of physical abuse in all infants with femur fractures is 1:2, and that a transverse fracture conformation carries an LR+ of 0.9, and that the presence of a co-occurring tibia fracture has an LR+ of 4.5, so the post-test odds are 2:1, and this is therefore a case where "Child abuse is likely", without asking if the caregiver reported abuse, or if the child had been witnessed to be injured after a stairway fall while carried by their parent, who landed on their leg. There are many other ways to apply Bayes' theorem that would be better, but a reader can't evaluate the use of Bayes' theorem with the information provided. Please give an example of how Bayes' theorem is included in the advice given to referring doctors.

- We have added an explanation in the Methods section, with a focus on the use of likelihood ratios. We do not state that we use Bayes theorem anymore, we explain how we use Bayesian reasoning to convert likelihood ratios to probabilities of abuse.

- The abstract is not sufficient to summarize the piece. Please include in the abstract that DECCA focuses on physical injuries. Please add more methods to the abstract: inclusion and exclusion criteria, the methods by which outcomes are measured (the abuse likelihood scale, how it was assigned, how satisfaction was measured) enrolment periods, survey methods - when/how/to whom was the survey distributed and how was a response encouraged?

- We have adapted the abstract, however the guidelines limit the abstract to 300 words. DECCA actually doesn't focus on physical injuries but on the interpretation of physical findings. This is also shown in the number of cases where we conclude that there is no indication of child abuse. Besides physical findings in a minority of cases we are consulted on other child abuse related findings. We hope that the reviewer finds our changes acceptable.

- Survey methods are inadequate. Please include the complete survey as an appendix. Please report survey methods (above). How was it determined that the DECCA advice "changed the perspective on the case"? What is the threshold for a changed perspective?

- We have added the following text 'This survey was specifically developed for this study and is available (in Dutch) upon request'. This survey is primarily aimed to assess the personal experience with DECCA and is not a validated survey.

- With respect to the second remark, this was literally the question for the advisee in the survey.

- Either give the data, or don't report the result. If you say that cases were referred "mainly during weekdays" or "mostly during office hours", you need to give numbers to back this up. Same for "...shifting over the years towards more requests from child safeguarding doctors."

- We have added the data on the referral pattern over the week. This as we feel that it underscores the importance of being accessible 24/7.

MINOR CONCERNS

- The authors' written English is certainly superior to my written Dutch - but the manuscript will require some extensive proof-reading for grammar.

- One of the collaborators of DECCA is native English speaker and she has cross-checked the manuscript.

- If 51.3% were boys, it is redundant to say that 48.7% were girls (but these percentages seem to exclude the 19 participants where sex was unknown).

- We thank the reviewer for noticing this error in our manuscript. Indeed in 19 children the sex was not reported/recorded. As sex is missing in a number of children we feel that is of importance to report both the number of boys and girls. We agree with the reviewer that if the sex was known in all cases this information could be considered to be redundant, but to be honest for the readers it is easier to read the number than to have to calculate them by hand.

- As I wrote before, the statistics in the paper are not well-chosen. The authors still report both median and mean for the same number. Rather than reporting every possible measure of central tendency and spread, the authors should choose the correct measure and report that one. For normally distributed data, this is the mean and standard deviation. For other data, this is generally the median and IQR.

- We have reported the median and IQR. The references to the medians have been removed from the manuscript. For the whole population the age range is also specified.

- Why the parentheses in the phrase "100% was (very) satisfied..."

- This means: were satisfied or very satisfied. The sentence is changed.

- The conclusions should summarize, rather than re-capitulate the results.

- We made changes to the conclusion.

- Strengths and limitations continue to report results, rather than report the *methodological* strengths and limitations of the study. Examples might include: Strengths - rather large sample with 4 years of data to show change. Limitations - relatively low survey response rate could be vulnerable to response bias (i.e., the least satisfied referrers might be least likely to fill out the survey - especially if not anonymous).

- We have made changes to the strengths and limitations section.

- I suggest that you search your document for the word "optimal" and replace it in every instance. To me, "optimal" implies - the best possible with no possibility of improvement. This is difficult to prove, and is not supported by these data. Similarly "despite its undeniable value" also seems to over-state the case.

- We have rephrased the two sentences in which the word 'optimal was used'.

- We have deleted "despite its undeniable value".

- When you say that the judgement of the injury is independent of social history, how can you ensure this? Keenan has shown implicit bias resulting from social data. Are experts blinded to social history? Is this even appropriate? After all, if you show me 2 children with identical fractures and tell me that one of them lives in a home with another abused child, I would reasonably believe that the risk of abuse is higher in that child.

- During the initial discussion between the between the paediatrician, who has spoken with the advisee, and the forensic physician the latter is blinded with respect to the social history and other risk factors. This way we aim to avoid cognitive bias as much as possible.

- I would not say (page 24) that geographic distribution was "skewed". Referrals came from the population centers, which is appropriate.

- We removed 'geographic distribution was "skewed"'. We intended to underline the fact that the majority of cases came from the west of The Netherlands.

- When you report that the most common injuries were "haematoma" - does this mean a bruise? subcutaneous hematoma? retroperitoneal hematoma? subdural?

- Indeed we refer to bruises here, we've changed this throughout the manuscript. Although most readers will use bruise and subcutaneous haemorrhage interchangeably we feel that bruise is more suitable in this context. Subdural hematoma are part of the category 'Brain damage'. Retroperitoneal hematoma are part of the category 'Internal injuries'.

Reviewer: 2

Name: Gabriel Otterman

Institution: Uppsala University Children's Hospital, Sweden Competing interests: None declared

Please leave your comments for the authors below Thank you for inviting me to review your manuscript entitled "The start of the Dutch Expertise Centre for Child Abuse; a descriptive study of the implementation and data of the first 4 years." I refer to my comments and author responses to reviewer and editorial comments to the submission from a year ago "bmjopen-2018-023608." The authors have included these discussions and responses in the current submission and have revised the manuscript with inclusion of an additional year of data as well as a helpful STROBE checklist.

The authors have carefully and systematically responded to address issues brought up in the comments to the 2018 submission in a way that has improved the manuscript to my satisfaction. In this renewed review I provide some suggested minor revisions for the authors to consider in order to further improve the manuscript for publication.

The authors have consulted a native English speaker for correction of minor spelling and grammar issues to consistently follow the Journal's style in UK English. The current manuscript contains some persistent style and grammar errors. A careful edit by a professional UK English editor is recommended. An example is the new title, where "start of" and "implementation" are perhaps redundant. Consider for example – "Implementation of the Dutch Expertise Centre for Child Abuse: descriptive data from the first four years." Other examples are use of "advices" in plural, inconsistent spelling of "hematoma / haematologist."

- Thank you for the suggestion of the title, we've decided to adopted this suggestion.
- We have re-checked the grammar of our manuscript and have used the Queen's English throughout the manuscript.
- One of the collaborators of DECCA is a native English speaker and she has cross-checked the manuscript as well.

In reference to the question by another peer review regarding the level of care (p10) that is not routinely recorded by DECCA, do the authors not consider this pertinent data? I'm thinking in particular about validated prediction tools for abusive head trauma evaluations such as PediBIRN which is correctly applied to the ICU population. Certainly, level of neurological impairment is an important data point for assigning likelihood ratios for this diagnosis. Given the low median age of DECCA referrals, abusive head trauma should be a common concern in this group (brain damage - Table 4).

- This data could indeed be of additional value but these data were not and could not be collected during the study period. The PediBIRN rule, although tested in a USA population has not been validated for use in a Dutch population. Cases of AHT form a minority of cases as these children will in general be transferred to one of the major paediatric university hospitals as the required neurosurgical care which is only available in these centres. These centres also have adequate paediatric care and expertise so DECCA advise is not routinely requested.

Despite the clarifications made by the authors in the new manuscript, I am still not clear on why the consultations with DECCA are anonymous. The authors cite confidentiality constraints such as GDPR as presenting a challenge to follow up of individual cases, as this would require obtaining consent

from the legal guardians / parents. I would argue that complete quality improvement cannot be secured without having the ability to follow up cases and outcomes of inter-agency investigations, child protective services and criminal justice over time.

- Because of anonymity for patient data, DECCA is easily accessible for a physician. Advice on a case can be asked without permission from a parent or caregiver. This is also in keeping with guidelines of the Royal Dutch Medical Association where in the first instance anonymous consultation is warranted. Although we agree that this leads to a loss of follow-up from a viewpoint of DECCA, that is of lesser importance compared to our advisory role.

There are clearly some exceptions in which DECCA recommends referral to an appropriate centre with expertise (p 9, line 18).

- This is indeed the case, however if a patient is referred to one of the three DECCA centres then this can only be done with parental consent. In these cases anonymity indeed is no longer an issue as they are part of normal paediatric care.

When DECCA provides a letter with an assigned likelihood of an abuse diagnosis, it is stated by DECCA that the letter should be included in the patient's medical record (p 9, line 1 □ p10, line 9). Are there not regulations that require the referring physician to obtain consent for the anonymized consultation?

- No, there is no regulation that requires the referring physician to obtain consent. However, in the majority of cases parents will be aware of the fact that the case will be discussed anonymously. In case of radiological review the parents have to consent as radiological revision is not anonymous.

Is the letter provided by DECCA documented in the medical record not made available to the patient/parent/guardian? If so, is not DECCA accountable to supervisory medical authorities as medical providers who have provided opinion on the record? Are DECCA providers not brought in to child protective or criminal justice investigations based on the provided opinion?

- The advisee is, after having been given advice from DECCA, primarily responsible for acting in the child's best interest. In case of child protective or criminal investigations, the advisee is in the lead, as DECCA is not able to provide any information because of the anonymity. We specifically state that our advice should be part of the patient's medical record and as such the parents have a right to read our letter. The Dutch system works different compared to many countries, where are hardly ever treating clinicians called into criminal court to testify. But if the judge wants to he/she can call anybody involved in the case so indeed the DECCA physicians could be asked to provide evidence in court.

An anonymised DECCA consultation letter may be illustrative if provided as supplemental material for this article.

- We don't see the additional value of adding an example to our manuscript. We have clearly described how we work and what is included in our letters.

I seek further clarification on these issues as it is a matter that we struggle with in our practice setting (Sweden) where we also a compelled to comply with GDPR regulations. It is my opinion that anonymising the consultation reduces the ability to ensure systematic improvement, and obstructs accountability in the system. These issues may thereby limit scalability of DECCA across countries and practice settings.

- All of our advisory requests and consultations are discussed in a weekly meeting and, and as a result of previous comments by this reviewer, as of this year we record if and how advice has changed after consultation. We therefore feel that we can ensure systematic improvement of the quality of our work. But the reviewer is correct that although this works in our legal system, there may be systems in the world where anonymity could be an obstruction. As we have no intricate knowledge

of all legal systems across the world, and it really is a legal and not a medical issue, we prefer not to add this to the discussion.

The authors state that inter-rater reliability of DECCA consultations has not been assessed, and yet assert that an advantage of the system is the assignment of a likelihood ratio for abuse versus non-abuse in accordance with a Bayesian analysis. Should not a quantified LR provide a ready means for inter-rater reliability? These could be assessed with the use of constructed case vignettes or anonymised cases.

- This is an interesting proposal for future research but this is not part of the method of DECCA.
- Additionally, as all cases are discussed during the weekly meeting, we have basically built in checks and balances to avoid erroneous conclusions.

The investigators state that the DECCA final advice provided in the consultation is “independent of social history” (p8, line 20). I understand that the experts may be striving to avoid cognitive biases that may be introduced through elements of social history. This does not however acknowledge the fact that referrals to DECCA are often triggered by the presence of social history concerns such as “presence of risk factors in family” (Table 3). I would argue that you cannot “unsee” the reason for the referral as a component that has bearing on the base rate for abuse in a Bayesian analysis.

- This is an interesting question, indeed there is a risk of a referral bias but during the first discussion between the paediatrician, who has spoken with the advisee, and the forensic physician these circumstances are not discussed. This way we aim to indeed avoid cognitive biases as much as possible. But the reviewer is correct, the mere fact that DECCA is consulted introduces a cognitive bias. We have made some minor changes in the text on the method of DECCA.

The authors risk overstating the case for DECCA by characterising it as “unique” (p 13, line 8, 16) or “optimal” expertise (p 16, line 13). These attributes have not been supported by the data.

- We have deleted the words ‘unique’ and ‘optimal’.

Finally, I'd like to restate that there is a need for programmes such as DECCA, especially as many paediatric colleagues, especially across Europe, seek to find ways to access reliable expertise in an expedited manner. A revised manuscript should be of compelling interest to many readers of BMJ Open.

VERSION 2 – REVIEW

REVIEWER	Daniel Lindberg University of Colorado
REVIEW RETURNED	19-Jun-2019

GENERAL COMMENTS	Thanks for the opportunity to review this revision. I feel that the paper has improved substantially, and have only minor suggestions.  - The finding that abuse was deemed to be unlikely in many cases is important - it counters the caricature of child abuse specialists as zealots who see abuse in every situation. Thank you for including this (no change suggested). - The prose has been much-improved in this draft and is easier to read. (no change suggested) - I very much appreciate the additional attention to the use of Bayes' theorem, but I still do not completely understand how this is done. I suggest that on page 9, a real-world example should be used after the explanation. I don't understand how information apart from the injuries is used to estimate a prior odds of abuse. Would this be something like, among all infants, the risk of
---

	physical abuse is 1:1000, or that among the poorest infants, the risk is 1:400?, and then rib fractures are more common in abused children (LR 9.4) so the post-test odds are 1:40? - "Paediatricians were the most prevalent advisees (53.1%), with child safeguarding doctors coming second." Please give the data for "child safeguarding doctors coming second" (22%) and/or cite Table 1. - "Most common injuries in advisory requests were bruises (264), fractures (166), skull fractures (69) and brain injury (64) (table 4)." Consider changing to "non-skull fractures (166), skull fractures (69)..." - Sorry to be "that guy" but my name is mis-spelled on page 14. (Linberg should be Lindberg). Thank you so much for citing my work.
--	---

REVIEWER	Gabriel Otterman Uppsala University Children's Hospital
REVIEW RETURNED	30-Jun-2019

GENERAL COMMENTS	Thank you for the request to review the revised manuscript. The authors have duly responded to the editorial and reviewer comments, and have made some significant improvements to the paper. Below are suggestions aimed to further improve the manuscript for publication. With all respect to the improvement of the manuscript following review by a native English speaker and the authors' use of Queen's English, there is a need for further proof-reading. An example is use of "western countries" (p 6) where "high-income countries" is preferred, or use of "manned" (p8) rather than the preferred "staffed." Another example is the use of "lays" (p 15) where "lies" is correct. Lindberg et al. are appropriately cited (p14) though Lindberg is misspelled. In response to reviewer comments regarding risks for overstatement, the authors have appropriately omitted the word "optimal," and have indicated in response that they omitted the term "unique." However, the word "unique" remains twice in the revision (p14) and should be excluded. It may be of interest to compare models of child abuse expert consultation that are being implemented elsewhere, such as the Kinderschutz Hotline in Germany. The authors helpfully describe some subgroups of patients that will not be made to DECCA (p13) as there is access to related expertise such as at the sexual assault centres. In the response to reviewer comments, the authors state that cases treated at the major paediatric university hospitals, e.g. abusive head trauma, clinicians have access to adequate paediatric expertise and DECCA advice would not routinely required. This description may be brought into the description in the paper to point out that there are other resources presumably available round the clock for some settings. On page 13, the authors state that in "most countries," hospitals are required to establish multidisciplinary teams. This was not what we found in a recent survey of child abuse paediatric professionals in Europe, where there was a majority of countries that reported having CPTs, but this was not necessarily due to any statutory requirement. If we think about middle and low-income
--

	countries, we are unlikely to find such a requirement for children's hospitals. The authors state that they "strongly believe" that the DECCA system increases the quality and consistency of the advice. The question is, rather, have they provided data to support this. It would be helpful instead to describe what steps are being undertaken to assure systematic improvement, such as the newly implemented recording of how DECCA advice has changed following case conferencing. What research is planned or needed in order to assess the model of forensic-paediatric collaboration and ensure quality? The discussion on page 16 regarding challenges with securing financing for DECCA is likely superfluous for the Journal's readers. While as practitioners we can empathise with the authors' challenges, this is not the correct forum for asserting that DECCA should be standard of care. The data which the investigators have presented effectively should speak for itself.
--	--

VERSION 2 – AUTHOR RESPONSE

Reviewer: 1

Reviewer Name: Daniel Lindberg

Thanks for the opportunity to review this revision. I feel that the paper has improved substantially, and have only minor suggestions.

- The finding that abuse was deemed to be unlikely in many cases is important - it counters the caricature of child abuse specialists as zealots who see abuse in every situation. Thank you for including this (no change suggested).

We feel that indeed this is as important as diagnosing child abuse.

- The prose has been much-improved in this draft and is easier to read. (no change suggested)

Thanks for these kind words.

- I very much appreciate the additional attention to the use of Bayes' theorem, but I still do not completely understand how this is done. I suggest that on page 9, a real-world example should be used after the explanation. I don't understand how information apart from the injuries is used to estimate a prior odds of abuse. Would this be something like, among all infants, the risk of physical abuse is 1:1000, or that among the poorest infants, the risk is 1:400?, and then rib fractures are more common in abused children (LR 9.4) so the post-test odds are 1:40?

This is an interesting comment. We feel that an example is not really fitting in the structure of the manuscript. However, we changed this paragraph and we believe that it should now be clearer to the readers of our manuscript.

- "Paediatricians were the most prevalent advisees (53.1%), with child safeguarding doctors coming second." Please give the data for "child safeguarding doctors coming second" (22%) and/or cite Table 1.

We've added a reference to table 1 at the end of this sentence.

- "Most common injuries in advisory requests were bruises (264), fractures (166), skull fractures (69) and brain injury (64) (table 4)." Consider changing to "non-skull fractures (166), skull fractures (69)..."

We have changed this throughout the manuscript, it will indeed make it clearer to the reader.

- Sorry to be "that guy" but my name is mis-spelled on page 14. (Linberg should be Lindberg). Thank you so much for citing my work.

No need to be sorry, we shouldn't have made this typo and of course we've corrected this.

Reviewer: 2

Reviewer Name: Gabriel Otterman

Thank you for the request to review the revised manuscript. The authors have duly responded to the editorial and reviewer comments, and have made some significant improvements to the paper. Below are suggestions aimed to further improve the manuscript for publication.

- With all respect to the improvement of the manuscript following review by a native English speaker and the authors' use of Queen's English, there is a need for further proof-reading. An example is use of "western countries" (p 6) where "high-income countries" is preferred, or use of "manned" (p8) rather than the preferred "staffed." Another example is the use of "lays" (p 15) where "lies" is correct. Lindberg et al. are appropriately cited (p14) though Lindberg is misspelled.

We do think that the first example is more a case of semantics than a question of grammar.

The use of high-income countries would also include a country like Japan, and we specifically intend to refer to Western European countries and North America. We did change this in our manuscript as it might confuse readers. In this sentence we refer to mandated reporting and this is what is published on Wikipedia 'The Council of Europe has urged all countries to have mandatory reporting of child abuse but several European countries do not. 15 member States (Bulgaria, Croatia, Denmark, Estonia, France, Hungary, Ireland, Lithuania, Luxembourg, Poland, Romania, Slovenia, Spain, Sweden and the United Kingdom) have reporting obligations in place for all professionals. In 10 member states (Austria, Belgium, Cyprus, the Czech Republic, Greece, Finland, Italy, Latvia, Portugal and Slovakia) existing obligations only address certain professional groups such as social workers or teachers'.

We've changed 'manned' into 'staffed'.

We've changed 'lays into 'lies'.

We've corrected the typo in the name Lindberg.

- In response to reviewer comments regarding risks for overstatement, the authors have appropriately omitted the word "optimal," and have indicated in response that they omitted the term "unique." However, the word "unique" remains twice in the revision (p14) and should be excluded. It may be of interest to compare models of child abuse expert consultation that are being implemented elsewhere, such as the Kinderschutz Hotline in Germany.

We are aware of the Kinderschutz Hotline which has been set up in Berlin, and now is as we understand expanding. However, the combination of a paediatrician discussing the case with a forensic doctor and report in a Bayesian way is to our knowledge unique. However, we've decided to remove the sentence 'To our knowledge this makes DECCA unique in the world' as we feel that removal doesn't essentially change the message of our manuscript.

On the second occasion where we used unique we've replaced this with 'excellent'.

- The authors helpfully describe some subgroups of patients that will not be made to DECCA (p13) as there is access to related expertise such as at the sexual assault centres. In the response to reviewer comments, the authors state that cases treated at the major paediatric university hospitals, e.g. abusive head trauma, clinicians have access to adequate paediatric expertise and DECCA advice would not routinely be required. This description may be brought into the description in the paper to point out that there are other resources presumably available round the clock for some settings.

Nowhere in the manuscript do we suggest that DECCA is involved in all cases, and the numbers shown in our paper clearly show this. We feel that it will be clear to the readers that paediatricians and other physicians can use other sources of information as well. Therefore we didn't change the manuscript.

- On page 13, the authors state that in "most countries," hospitals are required to establish multidisciplinary teams. This was not what we found in a recent survey of child abuse paediatric professionals in Europe, where there was a majority of countries that reported having CPTs, but this was not necessarily due to any statutory requirement. If we think about middle and low-income countries, we are unlikely to find such a requirement for children's hospitals.

We have changed this sentence into 'In many countries hospitals have some form of a hospital-based multidisciplinary team', we've also added a reference to the paper 'Paediatric approaches to child maltreatment are subject to wide organisational variations across Europe' which you co-authored. As we felt that this could be of interest to the readers of this manuscript.

- The authors state that they "strongly believe" that the DECCA system increases the quality and consistency of the advice. The question is, rather, have they provided data to support this. It would be helpful instead to describe what steps are being undertaken to assure systematic improvement, such as the newly implemented recording of how DECCA advice has changed following case conferencing. What research is planned or needed in order to assess the model of forensic-paediatric collaboration and ensure quality?

As discussed previously, due to the fact that we work with anonymous cases we cannot perform follow-up of the cases. Except for the routine questionnaires being sent to the advisees we, at this time, cannot do more research. The quality is ensured by our weekly telephone discussions and three monthly educational DECCA meetings.

- The discussion on page 16 regarding challenges with securing financing for DECCA is likely superfluous for the Journal's readers. While as practitioners we can empathise with the authors' challenges, this is not the correct forum for asserting that DECCA should be standard of care. The data which the investigators have presented effectively should speak for itself.

Although it may indeed be superfluous to some readers. However, for those who wish to start a similar program this is an important aspect of the process we've gone through. Therefore, and given the fact that the other reviewer didn't comment on this, we've decided to keep this in the manuscript.

We hope that with these responses in combination with the changes we've made to the text our manuscript now is acceptable for publication in BMJ Open.

VERSION 3 – REVIEW

REVIEWER	Daniel Lindberg University of Colorado School of Medicine
REVIEW RETURNED	08-Jul-2019

GENERAL COMMENTS	I still don't think the authors are being as clear as they think they are about how they use Bayes' theorem. I think reasonable reviewers, trying their best could still come up with 5 or 6 different ways that it could be applied which would produce drastically different interpretations (I know I can). I still think an example would help readers understand this central part of how DECCA works. But with all that said, the authors clearly disagree with me, and I don't think this concern is a deal-breaker. It's not worth holding up this paper for this issue, so I'm recommending acceptance.
--

REVIEWER	Gabriel Otterman Uppsala University Children's Hospital Sweden
REVIEW RETURNED	23-Jul-2019

GENERAL COMMENTS	The authors have done a terrific job of appreciably revising this manuscript to attentively respond to the reviewer comments. I commend the authors for their careful work on this revision and recommend publication in its current improved state. I am confident that the establishment and development of the DECCA Centre as described in this paper will be of compelling interest to the readers of BMJ Open.
---